# Cerebrospinal fluid dynamics correlate with neurogenic claudication in lumbar spinal stenosis

**Hyun-Ji Kim**[1], **Hakseung Kim**[1], **Young-Tak Kim**[1], **Chul-Ho Sohn**[2], **Keewon Kim**[3]*, **Dong-Joo Kim**[1,4]*

**1** Department of Brain and Cognitive Engineering, Korea University, Seoul, South Korea, **2** Department of Radiology, Seoul National University Hospital, College of Medicine, Seoul, South Korea, **3** Department of Rehabilitation Medicine, Seoul National University Hospital, College of Medicine, Seoul, South Korea, **4** Department of Neurology, Korea University Anam Hospital, Korea University College of Medicine, Seoul, South Korea

* dongjookim@korea.ac.kr (DJK); keewonkimm.d@gmail.com (KK)

## Abstract

Neurogenic claudication is a typical manifestation of lumbar spinal stenosis (LSS). However, its pathophysiology is still unclear. The severity of clinical symptoms has been shown not to correlate with the degree of structural stenosis. Altered cerebrospinal fluid (CSF) flow has been suggested as one of the causative factors of LSS. The objectives of this study were to compare CSF dynamics at the lumbosacral level between patients with LSS and healthy controls and to investigate whether CSF dynamics parameters explain symptom severity in LSS. Phase-contrast magnetic resonance imaging (PC-MRI) was conducted to measure CSF dynamics in 18 healthy controls and 9 patients with LSS. Cephalic peak, caudal peak, and peak-to-peak CSF velocities were evaluated at the lumbosacral level in the patients and controls. The power of CSF dynamics parameters to predict symptom severity was determined using a linear regression analysis adjusted for demographic and structural variables. Significantly attenuated CSF flow velocity was observed in the patients compared with the controls. The cephalic peak, caudal peak, and peak-to-peak velocities at the lumbar level were greater in the controls than in the patients ($p < 0.001$). The predictive power increased most when the peak-to-peak velocity was added (adjusted $R^2 = 0.410$) to the model with age, body mass index, and the minimum anterior-posterior diameter (adjusted $R^2 = 0.306$), and the peak-to-peak velocity was the only statistically significant variable. CSF dynamics variables showed an association with the severity of LSS symptoms, independent of structural stenosis. PC-MRI can help to further our understanding of the pathophysiology of neurogenic claudication and support the diagnosis of LSS.

## Introduction

Lumbar spinal stenosis (LSS) is a common degenerative disease characterized by a narrowed spinal canal in imaging and neurogenic claudication as a phenotype [1, 2]. Neurogenic

**Data Availability Statement:** All relevant data are within the paper and its Supporting information files.

**Funding:** HJK, YTK, HSK, DJK: This work was supported by the National Research Foundation of

Korea (NRF) grant funded by the Korea government (Ministry of Science and ICT, MSIT) [No. 2019R1A2C1003399, 2020R1C1C1006773]. The funders had no role in study design, data collection and analysis, decision to publish, or preparation of the manuscript.

**Competing interests:** The authors have declared that no competing interests exist.

claudication is characterized by unilateral or bilateral pain during walking, standing or lumbar extension [3]. Although LSS is highly prevalent and exhibits typical clinical manifestations, the pathophysiology of LSS remains unclear [4]. Mechanical nerve root compression has been proposed for the pathophysiology of neurogenic claudication. However, decompressive surgery to relieve stenosis does not always result in the effective relief of symptoms [5, 6]. Epidural steroid injections to relieve neural inflammation, which is another proposed pathogenesis of LSS, do not exhibit significant efficacy for relieving the pain of patients with LSS [3, 7]. Neurogenic claudication appears to be induced by vascular insufficiency [1, 8, 9]. Nonetheless, the administration of limaprost, a prostaglandin E1 analogue, to treat venous congestion also does not always yield satisfactory results [10]. LSS is clinically diagnosed based on a combination of the patient's clinical history, physical examination, and radiologic findings; a single pathognomonic test is not available because knowledge regarding the pathophysiology of this condition is insufficient. A narrowed spinal canal is a common radiological finding on spinal imaging of LSS patients; however, the extent of stenosis does not predict or explain the symptoms [11]. Structural abnormalities measured using conventional magnetic resonance imaging (MRI) or computed tomography do not always correlate with the functional state of patients with LSS [12], indicating that structural compression alone is not sufficient to explain symptom development.

Cerebrospinal fluid (CSF) dynamics have been proposed to be associated with neurogenic claudication [13]. Altered CSF dynamics in the brains of patients with hydrocephalus cause neurological deficits in the brain [14]. Likewise, altered CSF dynamics in the spine may also affect neural tissue in the spinal cord or nerve roots. Indeed, in LSS, patients with multilevel stenosis are more prone to develop neurogenic claudication than patients with single-level stenosis; dynamic rather than static structural components may contribute to the development of phenotypes [1, 15]. Changes in hydrostatic pressure caused by walking or changes in posture have been shown to be associated with neurogenic claudication [13, 16]. Forward bending may relieve claudication by increasing the spinal canal area, hence improving CSF circulation [1, 11, 13, 16]. A few previous studies have investigated the hydrodynamic parameters of CSF in the spinal canal of patients with spinal stenosis [13, 17]. In our pilot study, CSF flow was attenuated at the lumbosacral level in patients with LSS who experienced neurogenic claudication compared to normal healthy controls [18]. These various observations suggest that CSF dynamics may be associated with the pathophysiology of LSS.

The aim of this study was to examine the hypothesis that CSF dynamics independently contribute to the development of neurogenic claudication in LSS. For that purpose, we quantitatively measured CSF dynamics at the lumbosacral level using phase-contrast magnetic resonance imaging (PC-MRI) in patients with LSS and healthy controls and compared their hydrodynamic parameters. Furthermore, the predictive power of CSF dynamic variables for LSS symptom severity was evaluated in addition to epidemiologic and structural variables.

## Materials and methods

### Subjects

This level-3 investigational, observational study compared the CSF dynamics of patients with LSS and healthy controls. The study was approved by the institutional ethics committee of Seoul National University Hospital (Seoul, South Korea), and written informed consent was obtained from each subject (IRB number: H-1203-088-402). Patients with LSS were recruited from the outpatient clinic of a tertiary hospital and met the following criteria: (a) symptoms were experienced for more than 3 months; (b) claudication was provoked after less than 30 minutes of walking; and (c) spinal canal stenosis with a minimal anterior-posterior (AP)

diameter less than 10 mm was confirmed by lateral plain radiology. The exclusion criteria were as follows: (a) systemic peripheral neuropathy, peripheral vascular disease with vascular claudication, or a history of lumbar or brain surgery based on the patient's medical history; (b) segmental instability, which was defined as a > 4-mm translation of the sagittal plane or > 10˚ angular motion [19] based on dynamic plain radiograph, structural stenosis due to nondegenerative causes such as fractures or tumors, and foraminal stenosis that could explain the symptoms based on imaging studies; (c) concomitant myelopathy; or (d) a poor general medical condition that impeded gait function. Twelve LSS patients were initially enrolled in this study based on these inclusion and exclusion criteria. Subsequently, three subjects were excluded during the imaging analysis because the spinal canal was overlapped by aliasing artifacts from the kidney vessels. Thus, the LSS group consisted of nine patients, and the control group included eighteen healthy volunteers.

Demographic variables were obtained, including age, sex, and body mass index (BMI), which was calculated based on height and weight. A subjective assessment of symptom severity was conducted by recording the patient history and performing a physical examination. Patient history included information on the duration, characteristics, and distribution of symptoms and aggravating or relieving factors. Physical examination included the straight leg raise test, prone instability test, and neurological examinations, such as the manual muscle strength test, sensory examination, and deep tendon reflex. The ankle brachial index was assessed using a cuff manometer to exclude vascular claudication.

To evaluate symptom severity, all subjects completed the Oswestry disability index (scores range from 0 to 100 points) [20]. In this study, LSS symptom severity was evaluated based on 'claudication distance', which was defined as the distance one could walk without rest until restricted by the development of neurogenic claudication. Per the Oswestry disability index scale, the claudication distance was scored as 1 to 5 for walking distances less than 100 m, 100–499 m, 500–999 m, greater than 1000 m, and no claudication, respectively. A detailed description of the characteristics of the subjects is presented below (Table 1).

## Data acquisition

The MRI examinations were performed using a 3.0 Tesla MR system (Siemens, Magnetom Trio, Erlangen, Germany). The same PC-MRI protocol was used for all subjects (TR/

**Table 1. Baseline characteristics.**

| | Healthy controls (n = 18) | Patients with LSS (n = 9) | p-value[*] |
|---|---|---|---|
| Males, n (%) | 9 (50) | 4 (44.4) | 0.79 |
| Age, years | 56 (47–61.5) | 68 (57–75) | 0.015 |
| Body mass index, kg/m$^2$ | 24.2 (21.5–26.8) | 26.9 (25.6–29.8) | <0.001 |
| Oswestry disability index | 0 (0–1.25) | 18 (14–20) | <0.001 |
| Claudication distance score, n (%) | | | <0.001 |
| I | 0 (0) | 1 (11.1) | |
| II | 0 (0) | 3 (33.3) | |
| III | 0 (0) | 4 (44.4) | |
| IV | 0 (0) | 1 (11.1) | |
| V | 18 (100) | 0 (0) | |

[*] Categorical and continuous variables were assessed using the chi-square test and the Mann-Whitney *U*-test (t-test), respectively.

Data represented as median (interquartile range).

TE = 28.45 ms/9.47 ms, flip angle = 15˚, FOV = 320 mm, voxel size = 1×1×5 mm$^3$, slice thickness = 5 mm, 25% oversampling of the AP phase-encoding direction, bandwidth = 195 Hz/pixel, foothead single-direction velocity encoding, and generalized autocalibrating partially parallel acquisition with an acceleration factor of 2). T2-weighted images were acquired in the sagittal plane to determine structural stenosis.

CSF flow velocities were quantitatively evaluated at the level of the mid-vertebral body of the second lumbar vertebra (L2) and the first sacral vertebra (S1) using axial plane PC-MRI (Fig 1; and see Video in S1 Video, which illustrates the CSF flow). The following parameters were measured: the scan time was set based on the subject's heart rate averaged over 8 minutes (range 4–9 minutes), TR/TE = 34.60 ms/11.0 ms, flip angle = 15˚, FOV = 320 mm, voxel size = 1.3×1×7 mm$^3$, slice thickness = 7 mm, 60% oversampling of the AP phase-encoding direction, bandwidth = 130 Hz/pixel, through-plane single-direction velocity encoding, and no generalized autocalibrating partially parallel acquisition. Retrospective peripheral gating allowed 25 data points to be acquired over the entire cardiac cycle. A 4 cm/s velocity-encoding parameter was applied in all cases; velocity-encoding was determined to be 4 cm/s because the velocity of the CSF vertical flow at the lumbar or sacral level was observed to range between -2 cm/s and 2 cm/s based on a pilot study [18].

## Data analysis

The AP diameters of the canal were measured from the L1/2 to the L5/S1 mid-disc level using the mid-sagittal T2-weighted image. To indicate the severity of structural stenosis, the smallest AP diameter was designated as the minimum AP diameter (min AP). Region of interest (ROI) segmentation of the spinal canal was manually performed to measure CSF dynamics within the ventral part of the dural sac. An ROI was placed within the ventral part of the dural sac, showing an evident change in CSF flow on the axial phase images. Next, the location of the ROI was adjusted by excluding the dura mater or rootlets. The size of the rectangular ROI placed on the axial images was 0.3×0.3 cm$^2$. CSF flow velocity values were acquired for each of the 25-time frames over the entire cardiac cycle. The CSF flow velocity reveals the cephalic and caudal flow phases within one cardiac cycle: a positive velocity indicates the CSF flow upward along the axis (cephalic flow), and a negative velocity indicates the CSF flow downward along the axis (caudal flow). Three variables with simple but large descriptive power were calculated to represent CSF hydrodynamics: the maximum positive and minimum negative velocities were defined by the cephalic peak velocity and caudal peak velocity, respectively, and the peak-to-peak velocity was defined as the difference between the cephalic and caudal peak velocities.

## Statistical analysis

All statistical analyses were conducted with SPSS 25.0 (IBM Corp., Chicago, Illinois, USA). First, all the data were tested for normality using the Shapiro-Wilk test.

Parametric tests (t-test) were used to compare the normally distributed variables (BMI, min AP, cephalic peak velocity, caudal peak velocity, and peak-to-peak velocity), and nonparametric tests (Mann-Whitney $U$-test) were used to compare the categorical or discrete variables (sex, age, and claudication distance score) between the patients with LSS and the controls. The categorical variables, sex and claudication distance score, were examined using a chi-square test. A repeated-measures ANOVA was conducted to assess the variability in velocity at the sacral level during one cardiac cycle.

Associations of epidemiological, structural, and CSF dynamics variables with symptom severity were explored by performing a univariate linear regression analysis using the claudication distance score as the dependent variable. A multivariate linear regression analysis was

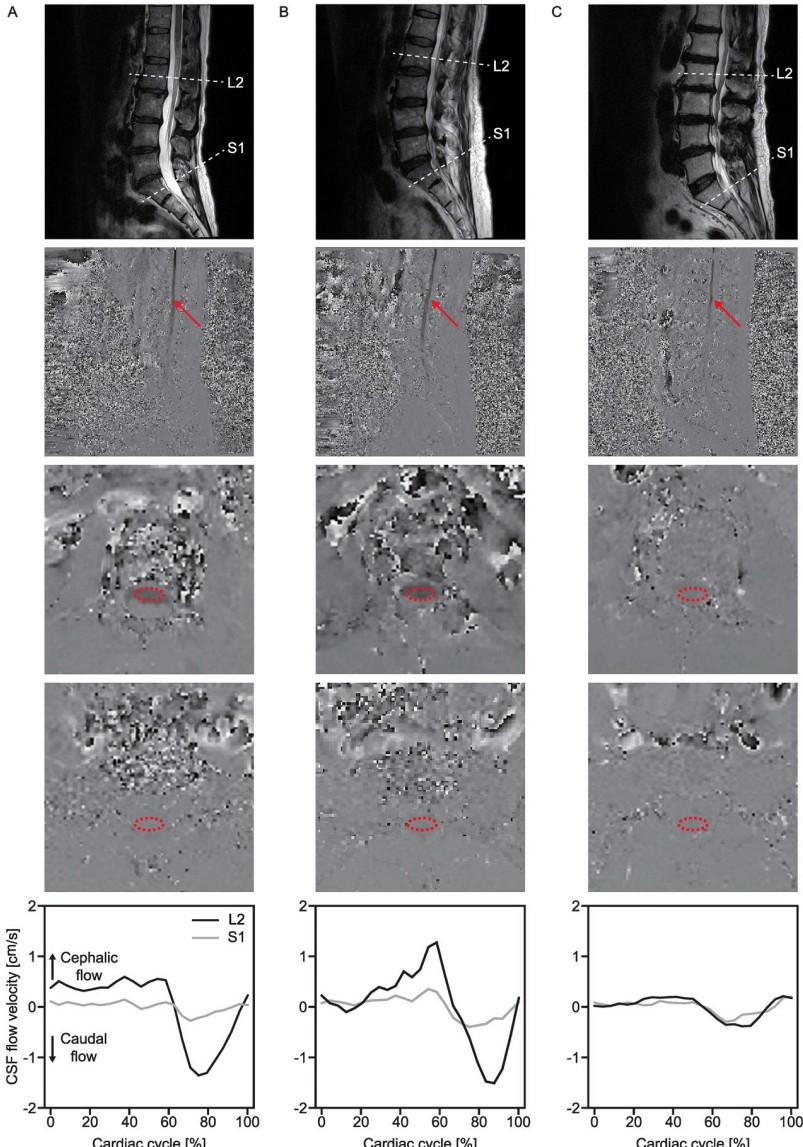

**Fig 1. Observations of CSF dynamics in three typical cases: A healthy control subject (A), a healthy control subject with a narrow canal (B), and a patient with lumbar spinal stenosis (C).** The 1st row depicts the T2-weighted MR images in the midsagittal plane, and the 2nd to 4th rows show the phase images in the midsagittal plane, L2 axial plane, and S1 axial plane, respectively. The 5th row shows the CSF flow velocity at the L2 (black line) and S1 (gray line) levels during one cardiac cycle. Subject A (a 56-year-old female) had no LSS symptoms and a minimum anterior-posterior diameter (min AP) of 10.38 mm at the L4/5 level. Subject B (a 61-year-old female) had no LSS symptoms but showed a narrow spinal canal with a min AP of 6.44 mm at the L4/5 level. Subject C (a 68-year-old male) had experienced neurogenic claudication for 1 year and showed multilevel stenosis at the L2/3, L3/4, and L4/5 levels with a min AP of 6.09 mm at the L4/5 level. A positive value indicates cephalic CSF flow, and a negative value denotes caudal CSF flow. The red arrowhead indicates CSF flow, and the red circle indicates the selected region of interest (ROI).

performed by adding or deleting epidemiological, structural, and hydrodynamic variables in multivariate models to determine whether the CSF hydrodynamic parameter contributed to symptom severity. Model 1 included the epidemiological variables, and structural variables were added to Model 1 to produce Model 2. Model 3 substituted CSF dynamics variables for the structural variables in Model 2. Model 4 included all the epidemiologic, structural, and

CSF dynamics variables. The variables in the multivariate models were those with a p-value <0.10 in the univariate analysis and were evaluated for the collinearity effect using the variance inflation factor. The adjusted R-square ($R^2$) value was calculated to evaluate the predictive power of each model. Statistical significance was set to a p-value of less than 0.05; a Bonferroni correction ($p < 0.017$ (0.05/3)) was applied to adjust for multiple pairwise comparisons between 3 groups.

## Results

### CSF velocity in patients with lumbar spinal stenosis compared with healthy controls

Considerable differences in CSF velocity at the lumbar level were observed between the patients with LSS and the healthy controls (Table 2 and Fig 2A). The patients with LSS showed significantly attenuated CSF velocities compared with the healthy controls (Table 2). At the sacral level, the maximum CSF velocity tended to be lower in the patients with LSS (healthy controls vs. patients with LSS, median (interquartile range): 0.6 (0.5–0.7) vs. 0.5 (0.4–0.6) cm/s, p = 0.03), but the velocity throughout the entire cardiac cycle was not significantly different from that of the healthy controls (p = 0.62; Fig 2B).

### Association of individual variables with symptom severity

A univariate linear regression analysis was conducted to determine the correlations between LSS symptom severity and demographic, structural, and CSF dynamics parameters (Table 3).

**Table 2. CSF flow velocity (cm/s) at L2 in the healthy controls and the patients with LSS.**

|  | Healthy controls | Patients with LSS | p-value* |
|---|---|---|---|
| Cephalic peak velocity | 0.8 (0.5–1.2) | 0.2 (0.2–0.3) | <0.001 |
| Caudal peak velocity | -1.3 (-2.0--0.9) | -0.4 (-0.5--0.3) | <0.001 |
| Peak-to-peak velocity | 1.9 (1.5–3.2) | 0.6 (0.5–0.7) | <0.001 |

* Based on the t-test.

Data represented as median (interquartile range).

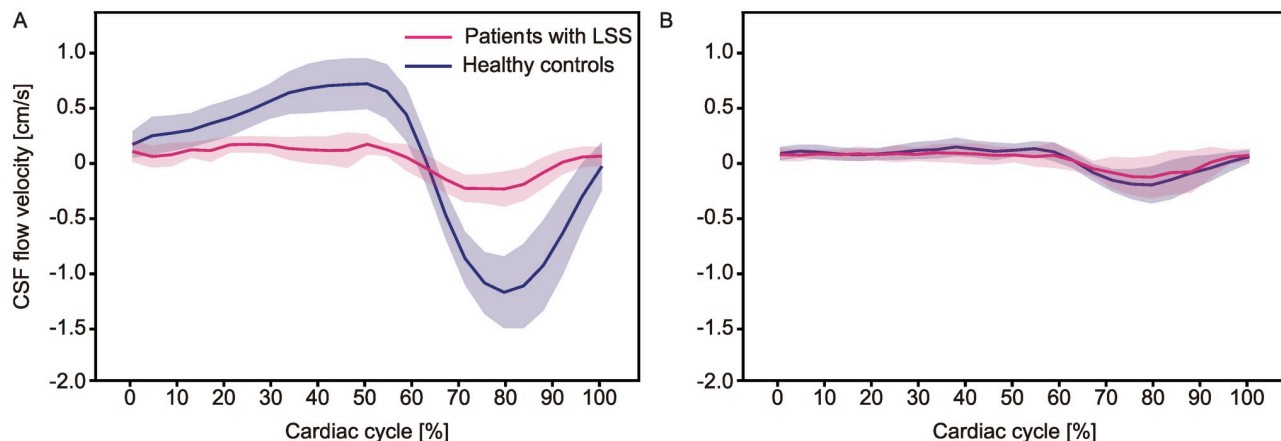

**Fig 2. Averaged CSF flow curves at the lumbosacral level over one cardiac cycle for patients with LSS and healthy controls.** CSF dynamics of a patient with LSS (pink line) and a healthy control (dark blue line) at L2 (A) and S1 (B). The lines represent the mean values and the shaded areas denote the 95% confidence interval.

**Table 3. Univariate linear regression analysis to explain the claudication distance score.**

| | B (95% CI) | Adjusted $R^2$ | p-value |
|---|---|---|---|
| Sex (male = 0, female = 1) | -0.385 (-1.401–0.632) | -0.015 | 0.44 |
| Age | -0.043 (-0.074–-0.012) | 0.214 | **0.009** |
| Body mass index | -0.155 (-0.290–-0.019) | 0.149 | **0.027** |
| Minimum AP diameter | 0.222 (0.044–0.400) | 0.178 | **0.016** |
| Cephalic peak velocity | 1.556 (0.678–2.434) | 0.321 | **0.001** |
| Caudal peak velocity | -1.036 (-1.586–-0.486) | 0.351 | **0.001** |
| Peak-to-peak velocity | 0.653 (0.311–0.995) | 0.358 | **0.001** |

Bolded: statistically significant.

Minimum AP diameter: minimum anterior-posterior diameter.

Except for sex, all the variables were significantly correlated with symptom severity as determined by claudication distance score: the claudication distance score decreased (more severe symptom) significantly with increasing (milder symptom) age, BMI, and caudal peak velocity and increased in subjects with a greater min AP, cephalic peak velocity, and peak-to-peak velocity.

## Predictive model for LSS symptom severity

Multivariate linear regression analyses were performed to determine whether CSF dynamics parameters independently contributed to symptom severity (Table 4). All the models were adjusted for age, BMI, min AP, and CSF dynamics variables. Peak-to-peak velocity, which was the CSF dynamics variable that exhibited the strongest $R^2$ value in univariate linear regression among the CSF dynamics parameters, was used as the CSF dynamics variable. Model 1, which included age and BMI, explained 26.2% of the variance in symptom severity. When the degree

**Table 4. Multivariate linear regression analysis to explain the claudication distance score.**

| | B (95% CI) | p-value | Adjusted $R^2$ |
|---|---|---|---|
| *Model 1 (demographic variables)* | | | |
| Age | **-0.034 (-0.066–-0.002)** | 0.038 | 0.262 |
| Body mass index | -0.105 (-0.240–0.029) | 0.12 | |
| *Model 2 (demographic + structural variables)* | | | |
| Age | **-0.032(-0.063–-0.001)** | 0.045 | 0.306 |
| Body mass index | -0.058(-0.202–0.087) | 0.42 | |
| Minimum AP diameter | 0.144(-0.043–0.330) | 0.13 | |
| *Model 3 (demographic + CSF dynamics variables)* | | | |
| Age | -0.015(-0.048–0.018) | 0.37 | 0.393 |
| Body mass index | -0.082(-0.206–0.042) | 0.19 | |
| Peak-to-peak velocity | **0.482(0.080–0.883)** | 0.021 | |
| *Model 4 (demographic + structural + CSF dynamics variables)* | | | |
| Age | -0.015(-0.048–0.018) | 0.35 | 0.410 |
| Body mass index | -0.047(-0.182–0.087) | 0.47 | |
| Minimum AP diameter | 0.11(-0.066–0.285) | 0.21 | |
| Peak-to-peak velocity | **0.437(0.034–0.840)** | 0.035 | |

Bolded: statistically significant.

Minimum AP diameter: minimum anterior-posterior diameter.

of stenosis was added to Model 1, the explanation of the variance increased to 30.6% (Model 2: age, BMI, and min AP). When the structural variables in Model 2 were replaced with the CSF hydrodynamics (Model 3: age, BMI, and peak-to-peak velocity), the descriptive power was found to increase markedly (adjusted $R^2 = 0.393$). Model 4, which included age, BMI, min AP, and peak-to-peak velocity, showed the highest goodness-of-fit and had peak-to-peak velocity as the only significant variable (adjusted $R^2 = 0.410$).

## Predictive model for LSS symptom severity in elderly subjects

Further linear regression analysis was conducted with closer age matching by excluding the younger healthy controls from all subjects (n = 23, mean age (range): 61.7 (49–76) years; S1 File and Table 5). The predictive power was increased by the addition of the CSF dynamics parameter (model 3 [age, BMI, and peak-to-peak velocity], adjusted $R^2 = 0.623$; model 4 [age, BMI, min AP, and peak-to-peak velocity], adjusted $R^2 = 0.609$). Age, which showed no significance in the model for all subjects, was significantly correlated with the claudication distance score in the model for aged subjects. Nevertheless, the predictive power of all the models showed improvements compared to the models analyzed in the all-subjects group (adjusted $R^2$ of the model with all subjects vs. elderly subjects, model 1: 0.262 vs. 0.359; model 2: 0.306 vs. 0.371; model 3: 0.393 vs. 0.623; model 4: 0.410 vs. 0.609).

## Discussion

This study was conducted to test whether CSF dynamics explained the severity of neurogenic claudication in LSS using PC-MRI. In the comparison of patients with LSS and healthy controls, greater CSF flow velocities were found in the healthy controls and the difference was particularly significant at the lumbar level. CSF dynamics parameters predicted the severity of LSS symptoms independent of epidemiological or structural parameters; the predictive power was improved by the addition of a CSF dynamics parameter.

**Table 5. Multivariate linear regression analysis to explain the claudication distance score in the elderly subjects.**

| | B (95% CI) | p-value | Adjusted $R^2$ |
|---|---|---|---|
| *Model A (demographic variables)* | | | |
| Age | **-0.087 (-0.145–-0.028)** | **0.006** | 0.359 |
| Body mass index | **-0.144 (-0.285–-0.003)** | **0.045** | |
| *Model B (demographic + structural variables)* | | | |
| Age | **-0.073 (-0.136–-0.011)** | **0.024** | 0.371 |
| Body mass index | -0.084 (-0.261–0.092) | 0.33 | |
| Minimum AP diameter | 0.146 (-0.115–0.406) | 0.26 | |
| *Model C (demographic + CSF dynamics variables)* | | | |
| Age | **-0.076 (-0.121–-0.030)** | **0.002** | 0.623 |
| Body mass index | **-0.113 (-0.222–-0.003)** | **0.045** | |
| Peak-to-peak velocity | **0.672 (0.309–1.036)** | **0.001** | |
| *Model D (demographic + structural + CSF dynamics variables)* | | | |
| Age | **-0.071 (-0.120–-0.021)** | **0.008** | 0.609 |
| Body mass index | -0.090 (-0.230–0.049) | 0.19 | |
| Minimum AP diameter | 0.0580 (-0.155–0.270) | 0.58 | |
| Peak-to-peak velocity | **0.647 (0.263–1.030)** | **0.002** | |

Bolded: statistically significant.

Minimum AP diameter: minimum anterior-posterior diameter.

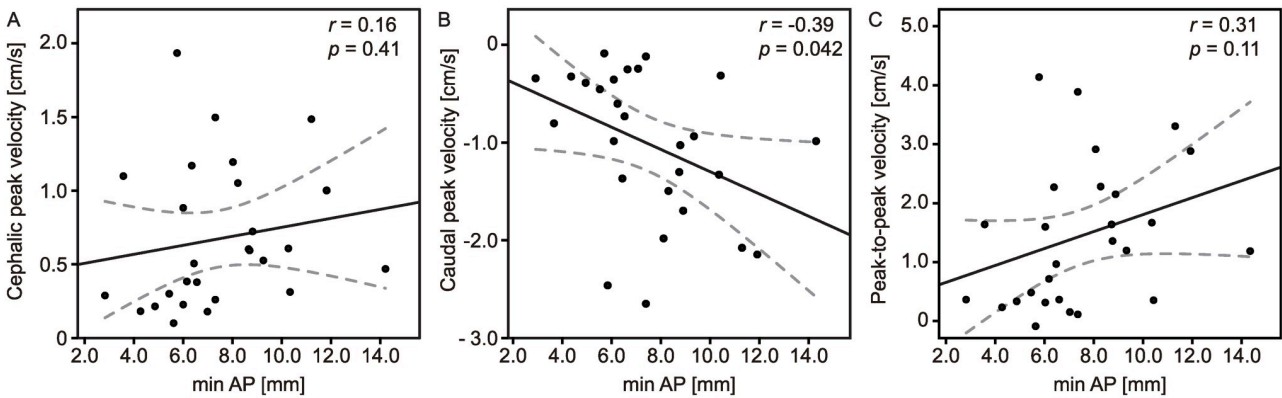

**Fig 3. Scatter plots of the min AP versus CSF dynamics variables.** Comparison of the min AP with the cephalic peak velocity (A), caudal peak velocity (B), and peak-to-peak velocity (C). Min AP = minimum anterior-posterior diameter.

Although CSF dynamics predict LSS symptoms, it remains possible that CSF dynamics are secondary to the reduced spinal canal area. We evaluated the association between CSF dynamics and minimum canal diameter. Except for caudal peak velocity (r = -0.394, p = 0.042), CSF dynamics variables, i.e., cephalic peak velocity and peak-to-peak velocity, had no correlation with the severity of stenosis (Fig 3). Furthermore, when we measured the cross-sectional area (CSA) at the lumbar mid-vertebral level on the axial images, the CSA in the LSS patients was significantly lower than that in the healthy controls (healthy controls vs. patients with LSS, median (interquartile range): 222.5 (195.0–255.3) vs. 180.0 (152.5–206.5) mm$^2$, p = 0.008). However, none of the CSF dynamics variables were correlated with the CSA (S1 Fig).

Another possible confounder that may entail the correlation between attenuated CSF dynamics and symptom severity is age. The age distribution in this study was uneven and could lead to biased results. To compensate for the uneven distribution of age, the healthy controls were divided into younger and older groups, and the CSF dynamics variables were compared between the subgroups (younger controls vs. older controls/younger controls vs. LSS patients/older controls vs. LSS patients). The CSF velocities measured at the lumbar level in the older control group tended to be lower than those in the younger control group, without statistical significance; however, those variables were remarkably decreased in the LSS patient group compared to both the younger and older controls (Fig 4 and Table 6). The reduced CSF dynamics at the lumbar level might be regarded as a specific observation in patients with LSS with neurogenic claudication. Nevertheless, further works should be conducted with a larger population to investigate the effects of age on the attenuation of CSF dynamics.

CSF dynamics in the spine are determined both by pulse generation from the neurovasculature of the brain and by flow resistance in the spinal canal. The attenuation of CSF pulsation might have resulted from reduced compliance of the central nervous system rather than from structural stenosis of the spine based on the Monro-Kellie doctrine [21]. A previous study of CSF flow in the spinal canal reported that spinal CSF pulsations are caused mainly by intracranial pulsations [22]. CSF pulsation at the cervical or cranial level has also been shown to decrease with age in accordance with reduced compliance of the brain as a degenerative process [23]. Symptomatic lumbar spinal stenosis can be considered a degenerative condition involving both the lumbar vertebral structure and the compliance of the CSF space.

Several previous studies have explained how changes in CSF dynamics affect neurogenic symptoms in patients with LSS. In terms of the kinetics of CSF, the theory of stagnant anoxia suggests that an increase in CSF pressure may block radicular venous drainage, resulting in

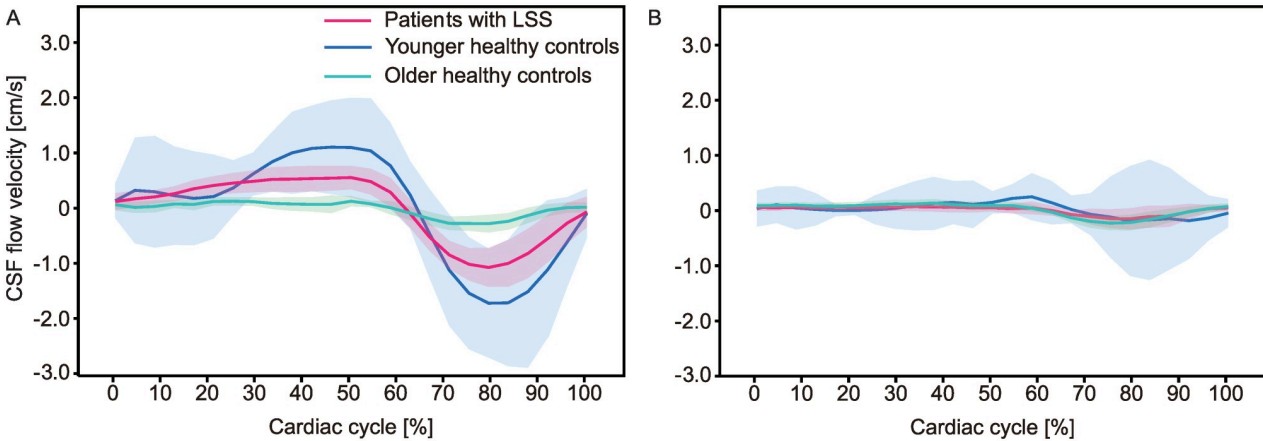

**Fig 4. The influence of age on the CSF flow was considered by separately comparing older (> 50 years old, light blue line) and younger (< 30 years old, blue line) healthy controls at L2 (A) and S1 (B).** The lines represent the mean values and the shaded areas denote the 95% confidence interval.

relative hypoxia or a reduction in metabolic exchange [24, 25]; however, attenuation of CSF pulsation would be related to the pressure gradient rather than the local pressure. Moreover, congested CSF circulation may disturb the supply of nutrients or oxygen to neural tissues [26]. During walking, the metabolic demands of the nerve root increase, while the oxygen or nutrient supply decreases [1], causing neurological claudication. In a previous pilot study, compared to normal healthy controls, patients with LSS who experienced neurogenic claudication showed attenuated CSF flow [18]. These changes may result in neurogenic claudication and explain how altered CSF dynamics contribute to neurological symptoms.

Although spinal imaging is considered the most informative examination for the diagnosis of LSS, its diagnostic value remains uncertain. In practical studies, there is a high prevalence of asymptomatic stenosis; approximately 20% of the elderly population exhibits asymptomatic spinal stenosis on spinal images [27, 28]. Nonetheless, an assessment of the status of CSF dynamics may improve the diagnostic accuracy of spinal imaging for LSS; for example, in the present study, the predictive power was increased when CSF dynamics variables were added to demographic and structural variables (Tables 4 and 5). In addition, when a logistic regression analysis was conducted to determine whether the addition of CSF dynamics parameters improved the ability to diagnose LSS, adding a CSF dynamics variable was more beneficial for predicting LSS than adding structural variables (chi-square value of model 1 [age, BMI]: 14.410, model 3 [age, BMI, peak-to-peak velocity] and 4 [age, BMI, min AP, peak-to-peak velocity]: 19.701, 1 degree of freedom; p < 0.001; Table in S2 File). Moreover, decompression

**Table 6. CSF flow velocity (cm/s) in the young and old healthy controls and patients with LSS.**

|  | Healthy control | | p-value* | Patients with LSS |
|---|---|---|---|---|
|  | **Young (n = 4)** | **Old (n = 14)** |  | **(n = 9)** |
| Age, years | 26 (25–28) | 58 (54–64) |  | 68 (57–75) |
| Cephalic peak velocity | 1.3 (1.0–1.8) | 0.6 (0.5–1.1) | 0.035 | 0.2 (0.2–0.3) |
| Caudal peak velocity | -2.1 (-2.4 --1.1) | -1.2 (-1.6--0.9) | 0.16 | -0.4 (-0.5--0.3) |
| Peak-to-peak velocity | 3.4 (2.2–4.2) | 1.9 (1.4–2.6) | 0.046 | 0.6 (0.5–0.7) |

* Adjustment for multiple comparisons: Bonferroni.

Data represented as median (interquartile range).

of structural stenosis does not guarantee clinical improvement in patients with LSS [5], probably because LSS symptoms are not only caused by compression but are also affected by altered neurophysiology within the CSF space. Therefore, evaluating CSF flow may improve the diagnostic accuracy of spinal imaging for LSS and support decision-making regarding treatment for LSS.

In this study, postprocessing of the ROI segmentation has limitations in assessing the CSF flow at the lumbosacral level. The ROI was obtained manually from the small area placed within the ventral part of the dural sac. The CSF flow within the ROI could be different from the total CSF flow in the spinal canal. However, as the spinal cord ends and divides into rootlets in the lumbosacral spine, it is difficult to delineate the CSF space. Moreover, the CSF flow at the lumbosacral level is much attenuated compared with that at the cervical level; it is hard to determine the adequate threshold of the dural sac. A previous study proposed a method for semiautomated ROI segmentation at the C2-C3 level by creating a parametric image with subsequent application of a visually selected threshold [29]. Future studies may implement an automated segmentation method for CSF flow at the lumbosacral level with adequate validation.

Several limitations of this study should be noted. First, kinetic parameters, such as CSF pressure, could not be analyzed using PC-MRI. The CSF pressure may more directly affect neurological impairments in LSS than CSF kinematics. Second, MRI-based tests cannot be used to evaluate CSF dynamics in an active state, which would have revealed pathophysiology more directly. Third, the patient group did not represent LSS patients in general because the current study included only patients with central stenosis without instability or lateral (foraminal) stenosis to test the effect of CSF dynamics within the canal on LSS symptoms under more homogenous conditions. Fourth, due to the technical constraints of PC-MRI, the detailed parameters of fluid dynamics (e.g., turbulent flow) were not considered. Future studies that employ computational fluid dynamics could model the kinetic parameters using PC-MRI. CSF flow in the brain or at the cervical level should be assessed, which would allow pulse generation to be directly measured. Furthermore, four-dimensional PC-MRI, which can be used to quantify velocities in the horizontal plane, will provide detailed insights into fluid dynamics. The main limitation of this study is the small cohort size, which may limit its statistical power for obtaining a convincing conclusion. A sufficient sample size could not be calculated, mainly due to the lack of relevant literature. Future research should test the present study's findings with larger sample sizes.

To conclude, attenuation of CSF flow at lumbar spine level is associated with development of neurogenic claudication in LSS, better than severity of structural stenosis. Evaluation of CSF dynamics using PC-MRI may help diagnose LSS in the clinic.

## Supporting information

**S1 Video. CSF flow in three typical cases: A healthy control subject (A), a healthy control subject with a narrow canal (B), and a patient with LSS (C).**
(MP4)

**S1 Fig. Scatter plots of the cross-sectional area versus CSF dynamics variables.**
(DOCX)

**S1 File. Univariate linear regression analysis to explain the claudication distance score in the elderly subjects.**
(DOCX)

**S2 File. -2 Log likelihood ratio test to determine the effect of factors on LSS.**
(DOCX)

**S3 File. ROC analysis of linear regression models to predict LSS.**
(DOCX)

**S1 Data. The relevant data about this manuscript.**
(XLSX)

## Author Contributions

**Conceptualization:** Keewon Kim, Dong-Joo Kim.

**Data curation:** Hyun-Ji Kim, Keewon Kim, Dong-Joo Kim.

**Formal analysis:** Hyun-Ji Kim, Keewon Kim.

**Funding acquisition:** Dong-Joo Kim.

**Investigation:** Hyun-Ji Kim.

**Methodology:** Hyun-Ji Kim, Chul-Ho Sohn.

**Supervision:** Keewon Kim, Dong-Joo Kim.

**Validation:** Hyun-Ji Kim, Hakseung Kim.

**Visualization:** Hyun-Ji Kim, Young-Tak Kim.

**Writing – original draft:** Hyun-Ji Kim.

**Writing – review & editing:** Hyun-Ji Kim, Hakseung Kim, Young-Tak Kim, Chul-Ho Sohn, Keewon Kim, Dong-Joo Kim.

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
