## [Decision Letter · Decision Letter 0]

12 Feb 2021

PONE-D-21-01349

Cerebrospinal fluid dynamics correlate with neurogenic claudication in lumbar spinal stenosis

PLOS ONE

Dear Dr. Kim,

Thank you for submitting your manuscript to PLOS ONE. After careful consideration, we feel that it has merit but does not fully meet PLOS ONE’s publication criteria as it currently stands. Therefore, we invite you to submit a revised version of the manuscript that addresses the points raised during the review process.

Please adress the suggestions for improvement from the Reviewers and amend your manuscript accordingly.

We look forward to receiving your revised manuscript.

Kind regards,

Michael C Burger, M.D.

Academic Editor

PLOS ONE

Journal Requirements:

Reviewers' comments:

Reviewer's Responses to Questions

**Comments to the Author**

1. Is the manuscript technically sound, and do the data support the conclusions?

Reviewer #1: Yes

Reviewer #2: Partly

2. Has the statistical analysis been performed appropriately and rigorously? 

Reviewer #1: Yes

Reviewer #2: I Don't Know

3. Have the authors made all data underlying the findings in their manuscript fully available?

Reviewer #1: No

Reviewer #2: No

4. Is the manuscript presented in an intelligible fashion and written in standard English?

Reviewer #1: Yes

Reviewer #2: Yes

5. Review Comments to the Author

Reviewer #1: The work presented is original and well presented, it could help clinicians working in this field.

Nevertheless some methodological points should be clarify before publication to improve the value of this paper.

Introduction : Authors should introduce previous pioneers works done on the CSF velocities in the spinal canal and discuss it

Origin of subarachnoid cerebrospinal fluid pulsations: a phase-contrast MR analysis. Magn Reson Imaging. 2000 May;18(4):387-95.

Henry-Feugeas MC et al

“Region of interest (ROI) segmentation of the spinal canal was manually performed to measure CSF dynamics within the ventral part of the dural sac. An ROI was placed within the ventral part of the dural sac, showing an evident change in CSF flow on the axial phase images. Next, the location of the ROI was adjusted by excluding the dura mater or rootlets. “

This is a crucial point of the paper; it should be interesting to see clearly the CSF ROIs obtained for the 3 cases presented in figure 1. We can see the CSF flow curves in the graph but we can’t see the Phase contrast images of the CSF flows around the spine!

Do the ROIs include all the CSF around the spine present in the slice?

Why a background of interest was not applied to avoid eddy current effects ?

It should also be important to know how many pixels were in the ROIs, what was the area of the ROIs ?

Discussion about MRI acquisition and the post processing steps is important to highlight that the measurements of the small CSF flows obtain in all the spinal canal should be interpreted with caution.

Figure 1,2,4 present CSF flow velocity. Is it the mean flow?

Please indicate on the graphs that positive values represent cephalic and negative caudal directions.

If these curves represent mean velocities of the entire CSF ROIs that mean that CSF have a net flow during the cardiac cycle. In S1 mainly in the cephalic direction whereas in S2 it is mainly in the caudal direction ! IMPOSSIBLE

If authors make the hypothesis that no net flow exist at this level and their ROIs include all the CSF around the spine, May be it should be an idea to automatically move all their CSF flow velocity curves on the Yaxis to correct the error of measurement and obtain a zero net flow. Then the results could be interpreted in a physiological way.

In the tables : simplify the numbers : example 0.7 cm/sec rather than 0.719 cm/sec unfortunately accuracy of the measurement is not so good !

Reviewer #2: This study provides an interesting look at spinal stenosis from the viewpoint of CSF dynamics that implements a PC-MRI technique to generate a more sensitive marker of neurologic claudication. The study compared CSF dynamics at the lumbosacral level between patients with LSS and healthy controls and investigated whether CSF dynamics parameters explain symptom severity in LSS. 18 healthy controls and 9 patients with LSS MRI examinations were performed.

However, the limited sample size and age distribution of cohorts that were compared severely limit the scope of analysis. In addition, the use of ANOVA and linear modeling may not be appropriate given the age distribution of the healthy group. These and other issues should be addressed to better validate use of PC-MRI in LSS.

Mayor concerns:

(1) The authors showed that there are non-negligible changes in PC-MRI measurements as a function of age; yet the LSS group is older than the control group and the control group is not normally distributed (with both young and old components). The linear models should be repeated with closer age matching, as univariate and multivariate linear models may be biased by bimodal distributions of variables.

(2) Line 141-142: The venc was selected as 4 cm/s because the CSF flow was between -2 and 2 cm/s. A reference is needed for this. Also, should different velocities be thought about in each group, if CSF dynamics are significantly altered in the LSS group?

(3) Table 2 – These velocities are measured in cross sections, so since the cross-sectional areas of the dural sac in each subject are not provided, it is not clear if this could be driving the differences seen in the velocity profiles.

(4) Line 316-317 – It can’t be said that CSF flow at the lumbar level explains the development of neurogenic claudication, but it is simply correlated with it. To explain it there has to be a definite physiological link, which, while hypothesized, has not been proven.

(5) It is stated in line 279-280 that increased CSF pressure leads to decreased metabolic exchange. Wouldn’t an increased CSF pressure be associated with higher peak amplitudes, the opposite of what is seen in this study?

(6) Table 4 – A log likelihood ratio test (LRT) would be more informative about the relative performance of each model in comparison.

(7) Should an ROC analysis be used for different models?

Other minor points:

(1) Line 43 – Define neurogenic claudication

(2) Line 45 – “Mechanical nerve root compression have been” should be “Mechanical nerve root compression has been”

(3) In Methods the authors stated that Bonferroni corrected p-values were accepted at p < 0.017, but in Tables 3 and 4 p-values were shown to be significant above this threshold. Were these p-values not subject to Bonferroni correction; explain why.

(4) Line 50 – Sentence doesn’t make sense grammatically

(5) Can you show an example of an ROI placed in the dural sac?

(6) Line 256-257 – This sentence doesn’t make sense

(7) Line 307 – Should be turbulent flow

(8) Fig 3 is intended to show that none of the CSF velocity profiles correlated with the severity of stenosis. There is no y labels for some plots.

6. PLOS authors have the option to publish the peer review history of their article (what does this mean?). If published, this will include your full peer review and any attached files.

Reviewer #1: No

Reviewer #2: No

---

## [Author Response · Author response to Decision Letter 0]

31 Mar 2021

The authors are grateful for the reviewers’ critical comments and the editor’s decision regarding the manuscript. We recognize the importance of the advice, and we have significantly revised the manuscript to the best of our ability.

The two reviewers shared concerns regarding the ROI segmentation of the spinal canal. Given the importance of these comments, we answer the overlapping comment first and then address the questions that were more specific. Furthermore, we also employed a professional academic editing service to correct any remaining grammar or syntax problems. We consider the revised manuscript to represent a significant improvement over the previously submitted version.

---

## [Decision Letter · Decision Letter 1]

13 Apr 2021

Cerebrospinal fluid dynamics correlate with neurogenic claudication in lumbar spinal stenosis

PONE-D-21-01349R1

Dear Dr. Kim,

We’re pleased to inform you that your manuscript has been judged scientifically suitable for publication and will be formally accepted for publication once it meets all outstanding technical requirements.

Kind regards,

Michael C Burger, M.D.

Academic Editor

PLOS ONE

Additional Editor Comments (optional):

Reviewers' comments:

Reviewer's Responses to Questions

**Comments to the Author**

1. If the authors have adequately addressed your comments raised in a previous round of review and you feel that this manuscript is now acceptable for publication, you may indicate that here to bypass the “Comments to the Author” section, enter your conflict of interest statement in the “Confidential to Editor” section, and submit your "Accept" recommendation.

Reviewer #1: All comments have been addressed

Reviewer #2: All comments have been addressed

2. Is the manuscript technically sound, and do the data support the conclusions?

Reviewer #1: Yes

Reviewer #2: Yes

3. Has the statistical analysis been performed appropriately and rigorously? 

Reviewer #1: Yes

Reviewer #2: Yes

4. Have the authors made all data underlying the findings in their manuscript fully available?

Reviewer #1: Yes

Reviewer #2: Yes

5. Is the manuscript presented in an intelligible fashion and written in standard English?

Reviewer #1: Yes

Reviewer #2: Yes

6. Review Comments to the Author

Reviewer #1: Authors have done an important work to respond to the reviewers comments

the paper have largely improve and present now coherent results.

finally the authors should increase the quality of the images presented because actually the quality doesn't allow to see clearly the rois and understand where the CSF was segmented.

Reviewer #2: All my previous comments have been properly addressed. I have no other critiques and I think the article could be an interesting contribution to the field

7. PLOS authors have the option to publish the peer review history of their article (what does this mean?). If published, this will include your full peer review and any attached files.

Reviewer #1: No

Reviewer #2: No

---

## [Editor Report · Acceptance letter]

15 Apr 2021

PONE-D-21-01349R1 

Cerebrospinal fluid dynamics correlate with neurogenic claudication in lumbar spinal stenosis 

Dear Dr. Kim:

I'm pleased to inform you that your manuscript has been deemed suitable for publication in PLOS ONE. Congratulations! Your manuscript is now with our production department. 

Kind regards, 

on behalf of

Dr. Michael C Burger 

Academic Editor

PLOS ONE